# Biochemical, Biophysical, and Structural Analysis of an Unusual DyP from the Extremophile *Deinococcus radiodurans*

**DOI:** 10.3390/molecules29020358

**Published:** 2024-01-11

**Authors:** Kelly Frade, Célia M. Silveira, Bruno A. Salgueiro, Sónia Mendes, Lígia O. Martins, Carlos Frazão, Smilja Todorovic, Elin Moe

**Affiliations:** Instituto de Tecnologia Química e Biológica António Xavier (ITQB-NOVA), Universidade Nova de Lisboa, Av. da Republica (EAN), 2780-157 Oeiras, Portugal; kellyfrade@itqb.unl.pt (K.F.); celiasilveira@itqb.unl.pt (C.M.S.); brunosalgueiro@itqb.unl.pt (B.A.S.); smendes@itqb.unl.pt (S.M.); lmartins@itqb.unl.pt (L.O.M.); frazao@itqb.unl.pt (C.F.); smilja@itqb.unl.pt (S.T.)

**Keywords:** heme protein, dye-decolorizing peroxidases, resonance Raman spectroscopy, X-ray crystallography

## Abstract

Dye-decolorizing peroxidases (DyPs) are heme proteins with distinct structural properties and substrate specificities compared to classical peroxidases. Here, we demonstrate that DyP from the extremely radiation-resistant bacterium *Deinococcus radiodurans* is, like some other homologues, inactive at physiological pH. Resonance Raman (RR) spectroscopy confirms that the heme is in a six-coordinated-low-spin (6cLS) state at pH 7.5 and is thus unable to bind hydrogen peroxide. At pH 4.0, the RR spectra of the enzyme reveal the co-existence of high-spin and low-spin heme states, which corroborates catalytic activity towards H_2_O_2_ detected at lower pH. A sequence alignment with other DyPs reveals that *Dr*DyP possesses a Methionine residue in position five in the highly conserved GXXDG motif. To analyze whether the presence of the Methionine is responsible for the lack of activity at high pH, this residue is substituted with a Glycine. UV-vis and RR spectroscopies reveal that the resulting *Dr*DyPM190G is also in a 6cLS spin state at pH 7.5, and thus the Methionine does not affect the activity of the protein. The crystal structures of *Dr*DyP and *Dr*DyPM190G, determined to 2.20 and 1.53 Å resolution, respectively, nevertheless reveal interesting insights. The high-resolution structure of *Dr*DyPM190G, obtained at pH 8.5, shows that one hydroxyl group and one water molecule are within hydrogen bonding distance to the heme and the catalytic Asparagine and Arginine. This strong ligand most likely prevents the binding of the H_2_O_2_ substrate, reinforcing questions about physiological substrates of this and other DyPs, and about the possible events that can trigger the removal of the hydroxyl group conferring catalytic activity to *Dr*DyP.

## 1. Introduction

The dye-decolorizing peroxidases (DyPs) or DyP-type peroxidases belong to the most recently identified family of heme peroxidases, which are unrelated to the classical peroxidases, showing different substrate affinities, structural divergence, and low sequence similarity [1,2]. The first identified member of this family, DyP from the fungus *Bjerkandera adusta* (*B. adusta*), was isolated and characterized in 1999 [3]. The enzyme showed no homology to any of the known peroxidases and was named after its ability to efficiently catalyze the decolorization of a broad range of industrial dyes [4]. Later, additional members were found in the proteomes of other fungi, as well as in several bacteria. Moreover, a new sequence analysis showed that these enzymes are prominent in bacteria, whereas only a small amount is found in fungi and higher eukaryotes [5,6].

Even though DyP’s physiological role and biological substrates still remain unclear, it is well established that they display different catalytic properties compared with classical peroxidases. They are active at acidic pH and show a broad substrate profile, including several classes of synthetic dyes (e.g., anthraquinone and azo dyes), metal ions (e.g., Mn^2+^), β-carotene, aromatic sulfides and phenolic or non-phenolic lignin-compound units [1,7]. The specificity constant values (*k*_cat_/*K*_m_) vary significantly among different DyPs, and some are inefficient peroxidases, which suggests that this activity might not always be relevant to their physiological roles [4]. Owing to the wide range of substrates that can be processed and some of their reaction characteristics, DyPs are considered promising biocatalysts, offering several advantages over classical peroxidases. The potential applications in lignocellulose hydrolysis (under acidic conditions) and in the detoxification of aromatic pollutants are among the most interesting [2,8,9]. DyPs have attracted interest for a variety of applications including decolorization of wastewaters derived from, e.g., the textile; cosmetic; pharmaceutical industries; development of bioelectrocatalytic devices for sensing or degradation of substrates of interest, such as hydrogen peroxide or persistent dye pollutants; bio-bleaching processes in pulp and paper industries; and lignin degradation and conversion to added-value products [2,3,5]. A lot of effort has also been put into engineering tailored enzymes with improved properties for biotechnological applications [10].

Structurally, DyP-type peroxidases also differ from the classical peroxidases. In contrast with the α-helix rich structures of the latter, DyPs are organized in two domains with a ferredoxin-like fold of four anti-parallel β-sheets surrounded by α-helices. The larger C-terminal domain houses the non-covalently bound heme *b* in a crevice between the two domains. Analysis at the level of protein sequences shows that DyPs possess highly conserved residues in the heme-binding site, including a characteristic GXXDG distal motif. The Aspartate replaces the distal histidine residue in classical peroxidases and is thought to take its role as an acid-base catalyst in the catalytic reaction. It is also part of a loop which connects the N-terminal and C-terminal domain of the DyPs ferredoxin domain [11]. Other conserved residues are a distal arginine, thought to be important for the coordination of hydrogen peroxide, or in some cases working as the acid-base catalyst, and a proximal histidine, the fifth ligand to the heme iron [1,6]. Based on primary sequences, the DyP-type peroxidases were originally classified into four subfamilies: A, B, C and D, with the bacterial enzymes constituting subfamilies A, B and C, and the fungal enzymes belonging to subfamily D [6]. In general, the B and C enzymes are predicted to be cytoplasmic and, consequently, are expected to be involved in intracellular metabolic pathways, while enzymes belonging to class A contain a TAT-dependent signal sequence, which suggests that they function outside of cytoplasm or extracellularly [12]. Later, based on tertiary structures, Yoshida and Sugano (2015) suggested a reclassification into classes P (primitive), I (intermediate), and V (advanced), in which class P consists of the previous class B, class I of the previous class A and class V of the previous classes C and D [13].

The catalytic cycle of DyPs was proposed to be the same as the well-characterized heme-peroxidases. It consists of a three-step reaction resulting in the oxidation of two substrate molecules and reduction of hydrogen peroxide to water [14]. In the first step, the resting-state ferric enzyme Fe(III) reacts with hydrogen peroxide to form a Compound I oxo-ferryl intermediate [Fe(IV) = O]^+.^ using two-electron oxidation, with hydrogen peroxide as an electron acceptor for cleavage. Subsequently, Compound I, oxidizes the electron donor substrate forming Compound II [Fe(IV) = O] and a substrate cation radical. Finally, Compound II oxidizes the substrate to return the protein to the native resting state, completing one catalytic cycle [15]. Despite the similar reaction characteristics, previous studies have shown that the mechanistic features of DyPs from diverse sources and subfamilies may be connected to their heme microenvironments and how it changes upon the formation of catalytic intermediaries [8,16,17,18]. The changes in the heme microenvironment originated with the formation of the catalytic intermediaries have been assessed using UV-visible and resonance Raman (RR) spectroscopy [8,16,17,18,19]. These are important tools to study the mechanism of the peroxidase catalytic cycle, including characterizing the reaction intermediates, measuring the rate of their formation and transformation, and the significance of amino acids in the active sites.

Recently, we expressed, purified, and performed an initial crystallographic analysis of a new bacterial DyP from the extreme-oxidative-stress-resistant *Deinococcus radiodurans* (*Dr*DyP) [20]. Based on the initial classification and the Peroxibase [21], *Dr*DyP belongs to the unexplored class C DyPs, which are organized within class V of the new categorization [20]. Our preliminary analysis of *Dr*DyP indicated that it possesses unusual properties as it contains an LS heme at physiological pH, suggesting a lack of peroxidase activity in these conditions [20]. Here, we have performed a biochemical, biophysical, and structural analysis of *Dr*DyP (native and mutant) to discover more in-depth information about the activity and specificity of the enzyme, as well as structural information about the heme cofactor.

## 2. Results and Discussion

### 2.1. Sequence and Activity Analysis

A sequence analysis of *Dr*DyP compared to four characterized and structurally determined members of the class V peroxidase family of DyPs (Appendix A), demonstrates that the proximal His heme ligand, and the four residues that are proposed to form the hydrogen peroxide binding pocket within this family of DyPs are conserved in *Dr*DyP (H324, D189, R348, L373 and F375). However, the so-called DyP fingerprint motif GXXDG has a substitution in the fifth position, containing a Methionine (M190) instead of a Glycine, as observed for the other DyPs. The sequence identity between *Dr*DyP and the previously characterized family V DyP peroxidases is in the order of 20%, and thus is not very high. However, since the residues that have been predicted to be important for heme coordination, substrate binding, and catalysis are conserved, it is likely that the activity of this DyP peroxidase is comparable to the other family V DyPs.

To assess the pH profile of the protein, the activity was determined using H_2_O_2_ and ABTS (2,2′-azino-bis(3-ethylbenzothiazoline-6-sulphonic acid) as substrates in Britton–Robinson buffer with a pH ranging from pH 1.5 to 5. The results showed that the enzyme has optimum activity for ABTS at pH 2.8 and almost no activity at a pH above 4.0 (Figure 1, left panel). The temperature-profile analysis was also conducted by performing activity measurements at different temperatures, ranging from 5 to 45 °C (Figure 1, right panel). The results showed that the enzyme exhibits optimal activity between 12 and 18 °C.

### 2.2. Buffer-Stability Analysis

The thermofluor-based study was performed in order to determine the enzyme’s stability in the purification and activity assay buffers. According to this, three different conditions were tested: (i) the buffer used for the protein purification, 0.05 M Tris-HCl pH 7.5, 0.150 M NaCl (reference), (ii) 0.1 M Citrate-Phosphate pH 3.0 (reference), and (iii) 0.1 M Britton–Robison pH 3.0 (used in activity assays). The interaction of the dye with the protein generates an experimental sigmoidal curve. Therefore, the data derived from the thermofluor experiments were analyzed taking into consideration the midpoint temperature of the protein-unfolding transition (*T*_M_—melting temperature), which reflects the temperature at which 50% protein is unfolded and bound to the fluorescent dye. The results confirmed the presence of a stable protein for the three different buffer systems tested in a temperature range between 4 and ~25 °C (Figure 2). However, and as expected, the reference buffer (pH 7.5) preserves a more stable protein in comparison to the buffers at pH 3.0 (*T*_M_ is higher in the former case). Within the buffer systems at pH 3.0, the best results were obtained with the Citrate-Phosphate Buffer, since a higher *T*_M_ and a better shaped curve is observed at the beginning of the experiment, with a lower SYPRO Orange fluorescence. The level of the baseline in the three conditions tested is justified using some hydrophobic residues present at the protein’s molecular surface. However, the higher SYPRO Orange fluorescence at the beginning of the experiment for the Britton–Robison buffer at pH 3.0 suggests a mixture of a folded and unfolded protein in solution.

### 2.3. Structural Analysis of Heme in DrDyp and DrDypM190G

*Dr*DyP’s heme active site was previously characterized in detail using resonance Raman (RR) spectroscopy employing Soret band excitation (413 nm laser) [19]. In these conditions, the high-frequency region spectra (1300–1700 cm^−1^) of heme proteins reveal ‘core-size’ marker bands sensitive to the redox, coordination, and spin state of the heme iron. It was demonstrated that the ferric protein adopts a hexacoordinate low-spin (6cLS) state at pH 7.5. In this case, the protein is catalytically incompetent as it is unable to bind to H_2_O_2_ and form high-valence ferryl intermediates (Compounds I and II). The RR spectrum obtained at pH 4.0 revealed a formation of 6cHS ferric heme together with the 6cLS species. The relative amount of the HS population increases as the pH decreases, which agrees with the increased peroxidase activity observed below pH 4. Considering ν_4_ mode relative intensities, the pH-dependent 6cHS population represents 10–15% of the protein sample at pH 4.0 and about 30% at pH 3.0 (Appendix A). The activation of *Dr*DyP in a lower pH range was also supported by UV-Vis spectra obtained in the presence of H_2_O_2_ [20]. At pH 3.0, slight changes are observed in the spectrum, namely the downshifting of the Soret (413 nm) and charge-transfer (CT) (639 nm) bands to 417 and 655 nm, respectively, possibly due to the formation of a peroxidase catalytic intermediate, which we tentatively assign to Compound II, as in *Bs*DyP [16,22]. Note that no electronic-absorption high-spin (HS)-state markers (CT bands) are observed in *Dr*DyP’s spectrum above pH 4.0 [20].

As described earlier, the only residue important for substrate binding that is not conserved in *Dr*DyP is Met190 in the GXXDG motif. The proximity of this residue to the catalytic residue Asp189 made us suspect that it might affect the activity of the protein; we, thus, re-engineered a typical DyP distal site by generating the *Dr*DyPM190G mutant for spectroscopic analysis.

The UV-Vis spectra of *Dr*DyPM190G are similar to the native protein (Figure 3, top panel) [20]. They suggest the presence of a LS heme, with Q bands at 537 and 576 nm. At low solution pH (4.0), this LS heme co-exists with an HS species, characterized by Q and CT bands at 500 and 626 nm, respectively. Despite the substantially blue-shifted CT band in comparison with the native *Dr*DyP (639 nm) [20], we have also assigned it to a 6cHS heme species, since CT bands have been shown to range from 600 to 637 nm for this spin state [23]. RR spectroscopy of *Dr*DyPM190G confirms the presence of the same 6cLS and 6cHS heme states identified in *Dr*DyP (Figure 3, middle and bottom panels). The band-fitting analysis, in which the experimental spectra were deconvoluted into component spectra, was performed with almost identical band positions and line-widths used for the native protein [19]. This indicates that the replacement of the Met190 residue does not change the heme iron environment, excluding this residue as a possible 6th axial ligand of the heme iron. It is noteworthy that conversion to the 6cHS species appears to be facilitated in *Dr*DyPM190G, since a residual HS population can be detected at pH 7.5 (2–5% of total ν_4_ intensity) and over 25% of the protein is converted to an HS state at pH 4.0 (Figure 3).

### 2.4. Redox Properties of DrDyP

Despite not being directly involved in the catalytic cycle, the reduction potential of the Fe^3+^/Fe^2+^ transition in heme peroxidases is considered a good indicator of the redox properties of the catalytically relevant intermediates. Herein, we have determined formal reduction potentials, E°′ of *Dr*DyP, and *Dr*DyPM190G using cyclic voltammetry (CV) and pyrolytic graphite working electrodes. The CV signal of *Dr*DyP shows a pair of current peaks, which can be ascribed to the one-electron oxidation/reduction of the heme iron (Figure 4, top panel). The anodic and cathodic peak-currents ratio is close to one, and both are directly proportional to the scan rates (10–500 mV/s), as expected for a diffusionless electrochemical process (Appendix A). The E°′ of *Dr*DyP and *Dr*DyPM190G are found to be –81 ± 4 mV and −96 ± 7 mV, respectively, at pH 7.0, which we attribute to the 6cLS heme species, which is the major population present in the solution in both cases (cf. RR section). The overall signal intensity is considerably lower in *Dr*DyPM190G compared to *Dr*DyP, despite the same amount of protein being deposited on the electrode, which suggests that *Dr*DyPM190G may be less stable. Its E°′ is slightly more negative than that of *Dr*DyP, which may be due to increased solvent accessibility to the heme pocket, because of the removal of Methionine’s bulky side-chain.

The reduction potential of *Dr*DyP is within the range of values reported for other DyPs, −40 to −320 mV [8,18,22,24,25]. Peroxidases typically display negative reduction potentials, as this should assure that the ferric form is stable under physiological conditions, and, therefore, readily available to be oxidized using hydrogen peroxide to initiate the catalytic cycle. Like *Bs*DyP, *Dr*DyP’s reduction potential is in the moderately high range, which along with the high amount of the catalytically inactive 6cLS heme iron may explain its poor performance as a peroxidase.

### 2.5. Crystal-Structure Determination and Analysis 

To gain more insight into the function of DyP from *D. radiodurans*, both *Dr*DyP and *Dr*DyPM190G were crystallized and structurally determined to 2.20 and 1.53 Å resolution, respectively. The data collection and refinement statistics are shown in Table 1. *Dr*DyP possesses one and *Dr*DyPM190G possesses two molecules in the asymmetric unit, respectively. The final model of *Dr*DyP (residue 19–460) contains one heme molecule, two glycerols, three Magnesium ions and 45 waters. Each molecule of the *Dr*DyPM190G (residue 15–460) contains one heme molecule, one calcium (Ca), one Cl, one hydroxyl (OH), and 632 waters.

The overall structures of the proteins are similar to previously determined structures of family V DyP homologs from *Bjerkandera adusta*, *Auricularia-judea*, *Irpex lacteus* and *Pleurotus ostreatus* [26,27,28,29], with an irregular elliptic shape containing a ferredoxin-like fold consisting of an antiparallel beta-sheet (β1–β4) and two alpha helices (α1 and α6) in the N-terminal domain of the protein (Figure 5, left panel). In total, *Dr*DyP contain 12 α-helices and 10 β-strands, with the β-sheets located mainly in the interior of the protein and the α-helices on the surface of the protein.

A superimposition of the two structures demonstrates that they are almost identical with an RMSD value of 0.28 (Appendix A). The assumed catalytic residue in both structures (Asp189 and Arg248) and the axial His324 are well-aligned relative to the heme (Figure 5, top right panel). Interestingly, we could observe one hydroxyl group and one water molecule within the hydrogen bonding distance to the iron and the catalytic Asp189 and Arg248 on the distal side of the heme for *Dr*DyPM190 (Figure 5, bottom right panel). This was not observed for the *Dr*DyP structure and can most likely be explained with the lower resolution of this structure. The presence of water molecules on the distal side of the heme have previously been shown in crystal structures of other DyPs (reviewed in [30]). Here, we suggest that the presence of the hydroxyl group and the water molecule at the distal side of the heme iron can explain the hexacoordinated low-spin (6cLS) state of the heme of *Dr*DyP, and thus the lack of activity of *Dr*DyP at physiological pH.

In conclusion, we demonstrate, here, insights into structural and biochemical features of a DyP from the extremophile *Deinococcus radiodurans*. *Dr*DyP is inactive at physiological pH, due to the fact that the heme adopts 6cLS state at pH 7.5, as shown using the resonance Raman experiments. The high-resolution crystal structure of *Dr*DyPM190G revealed the presence of one hydroxyl group and one water molecule within hydrogen bonding distance of the heme and the catalytic Aspartate, blocking the access of H_2_O_2_ to the active site. We observe a tendency for formation of catalytically competent HS species as the pH decreases. However, one can speculate that in vivo; other key players or processes such as interactions with other proteins (e.g., encapsulin) govern *Dr*DyP’s properties and activity.

## 3. Materials and Methods

### 3.1. Recombinant Protein Production

The gene encoding of a three-amino-acid N-terminally truncated version (∆3) of DR_A0145, accession code Q9RZ08, hereafter called *Dr*DyP, from *Deinococcus radiodurans* was cloned into the pDest14 expression vector from the Gateway system (GE-healthcare) as previously performed for Endonuclease III [31], and using the primers described previously [20]. The primers contain nucleotides encoding for an N-terminal HisTag followed by a TEV cleavage site upstream of the gene encoding the protein. The mutant *Dr*DyPM190G was constructed using the QuikChange^®^ site-directed mutagenesis kit according to the manufacturer’s instruction (Stratagene), using pDest14 with the gene encoding *Dr*DyP as a template and the following primers: Fp*Dr*DyPM90G 5′-CTG GGC TAC AAG GAC GGC ATC AGC AAC CCG GCC-3′ and RP*Dr*DyPM90G 5′-GGC CGG GTT GCT GAT GCC GTC CCT GTA GCC CAG-3′. The insertion of the gene in the correct reading frame of the pDest14 vector and the mutation was confirmed using the GATC biotech sanger sequencing service (GATC biotech).

Expression plasmids containing the gene encoding *Dr*DyP and *Dr*DyPM190G were transformed into BL21(DE3)* and expressed on a large scale in LB-medium with a 200 μg/mL Amp at 37 °C. The cells were harvested after three hours of induction (initiated with addition of 0.5 mM IPTG), and were dissolved in 20 mL of extraction buffer (50 mM Tris-HCl pH 7.5, 150 mM NaCl, 10 mM MgCl_2_, 1 tablet EDTA free proteinase inhibitor from Roche and dNase from Merck KGaA, Darmstadt, Germany, followed by four freeze/thaw cycles in liquid nitrogen and a water bath at 22 °C. The resulting extract was centrifuged at 18,000 rpm at 4 °C and the soluble fraction was collected. The purification was performed as described previously [20]. In short, the soluble fraction was applied to a 1 mL HisTrap column (GE-healthcare, Chicago, IL, USA) equilibrated in buffer A (50 mM Tris-HCl pH 7.5, 150 mM NaCl). The protein was eluted over a gradient of 20 CV buffer B (50 mM Tris-HCl pH 7.5, 150 mM NaCl and 500 mM Imidazole). The red/brown coloured fractions containing *Dr*DyP were pooled and dialyzed overnight (4 °C) into buffer A. A HisTagged TEV protease [32] was added to the sample before dialysis (in the proportion of 15 mg protein to 1 mg TEV). After dialysis, the protein solution was applied to a second HisTap column and collected from the flow through. The pure protein was concentrated to about 10 mg/mL and stored at 4 °C. The purity was analyzed using SDS-PAGE and estimated to be greater than 95%.

### 3.2. UV-Visible and Resonance Raman Spectroscopy

UV-Visible and RR experiments were performed using 2–4 μM and 150–300 μM *Dr*DyP or *Dr*DyPM190G, respectively, in 40 mM Britton–Robinson (BR) buffers at different pH values (3–10). UV-Visible spectra were recorded using a Shimadzu UV-1800 spectrophotometer (Shimadzu, Kyoto, Japan) with a temperature-control module set to 18 °C. Formation of *Dr*DyP reaction intermediates was monitored after the addition of excess H_2_O_2_ (0.04–4 mM) to the protein solutions. RR solution measurements were completed at room temperature (RT) with 413 nm excitation source. A rotating quartz cell (Hellma, Müllheim, Germany) containing ca. 90 μL of sample was used in all measurements. The spectra were collected in backscattering geometry using a confocal microscope equipped with an Olympus 20× objective (working distance of 21 mm, numeric aperture of 0.35). The microscope was coupled to a Raman spectrometer (Jobin Yvon U1000, Edison, NJ, USA) with a 1200 lines/mm grating and a liquid-nitrogen-cooled CCD detector (Horiba). The laser beam was focused onto the sample with a power of 1.5–3.0 mW and 40–60 s accumulation time. Typically, 4–10 spectra were co-added in each measurement to improve signal-to-noise ratio. RR spectra were subjected to polynomial baseline subtraction and component analysis as described previously [33].

### 3.3. Electrochemistry

Cyclic voltammetry experiments were performed using a Princeton Applied Research 263A potentiostat. Measurements were conducted in a three-electrode cell composed of a working pyrolytic graphite electrode (edge plane; ∅3 mm; home-made with graphite from GE Healthcare), a platinum-wire counter electrode (Radiometer) and a Ag/AgCl (3 M, KCl) reference electrode (WPI). The working electrode was polished with alumina slurry (0.3 μm particle size; Buehler), thoroughly washed with water, ultrasonicated for 5 min, and dried with a N_2_ flow. Then, a small droplet of the protein solution (~3 μL of 100 μM *Dr*DyP or *Dr*DyPM190G mutant in 50 mM Tris-HCl, pH 7.5) was deposited on a freshly clean electrode surface and left drying at RT for ca. 15 min. Control electrodes (without protein) were prepared by depositing a small droplet of protein buffer on the electrode surface. Afterwards, the electrode was rinsed with supporting electrolyte (0.05 M KCl in 40 mM BR buffer at pH 7.5) and placed in the electrochemical cell. The supporting electrolyte was thoroughly purged with argon before each experiment. The potential was cycled between 0.1 and −0.7 V vs. reference electrode at scan rates between 10 and 500 mV s^−1^. Potentials are quoted vs. the normal hydrogen electrode (NHE, 0.21 V vs. Ag/AgCl electrode).

### 3.4. Biochemical Assays

The enzymatic activity of *Dr*DyP was monitored using either a Nicolet Evolution 300 spectrophotometer (Thermo Industries, Madison, WI, USA), or a Synergy2 microplate reader (BioTek, Winooski, VT, USA). All enzymatic assays were performed at least in triplicate. The activity dependence on pH was measured by monitoring the oxidation of 1 mM of 2,2-azino(3-ethylbenzothiazoline-6-sulfonic acid); ABTS at 420 nm (ε420 nm = 36,000 M^−1^cm^−1^), in the presence of 1 mM H_2_O_2_ at 20 °C using Britton–Robinson (BR) buffer (0.1 M phosphoric acid, 0.1 M boric acid and 0.1 M acetic acid mixed with NaOH to the desired pH; pH in the range of 1.4–11). The temperature profile (between 5 and 45 °C) was determined in 0.1 M phosphate-citrate buffer at pH 2.8 by monitoring 1 mM ABTS oxidation at 420 nm in the presence of 1 mM H_2_O_2_.

### 3.5. Buffer Stability Analysis (Thermofluor)

The thermal-shift assay was performed on an iCycler iQ5 real-time PCR-detection system (Bio-Rad, Hercules, CA, USA), equipped with a charge-coupled device (CCD) camera and a Cy3 filter with excitation and emission wavelengths of 490 nm and 575 nm, respectively. The method used was described previously by Santos et al., 2012, and Kozal et al., 2016 [34,35]. In short, the PCR microplate (Bio-Rad, Hercules, California, United States) was prefilled with 18 μL of solution per well from the fresh buffer solutions: 0.1 M Citrate-Phosphate pH 3.0; 0.1 M Britton–Robinson pH 3.0; and 0.05 M Tris-HCl pH 7.5/0.150 M NaCl (protein purification buffer). Then, 2 μL of a master mix solution was added to each well. The solution was composed of the purified *Dr*DyP (12.5 μg/μL and SYPRO Orange dye (Invitrogen Molecular ProbesTM), both diluted from the initial 25 μg/μL and 5000-fold stock, respectively, in 0.05 M HEPES pH 8.0). Thus, the master mix solution was diluted 10× in the experimental buffers. The microplate was sealed with an Optical Quality sealing tape (Bio-Rad, Hercules, CA, USA) and centrifuged at 2500× *g* for 2 min, immediately before starting the assay, to remove possible air bubbles. The fluorescence was measured at regular intervals with a temperature gradient of 1 °C per minute and a 10 s hold step for every point for a temperature range between 4 to 90 °C.

### 3.6. Crystallization, Data Collection, Processing, Structure Determination, and Refinement

The crystallization condition, data collection, and processing of *Dr*DyP was described previously [20]. The crystallization of *Dr*DyPM190G was performed as follows. A protein solution of 10 mg/mL was used in sitting drop crystallization experiments using a Honeybee Cartesian crystallization robot (Genomics systems) with MDL3 plates using 100 nl protein and 40 mL reservoir solution and the Morpheus crystallization screen (Molecular Dimensions, Calibre Scientific, Rotherham, UK). Small crystals appeared in condition A9 with a reservoir solution consisting of 0.06 M Divalents (0.3 M Magnesium chloride hexahydrate; 0.3 M Calcium chloride dehydrate), 0.1 M Buffer System 3 (1.0 M Tris (base); BICINE, pH 8.5), and 50% *v*/*v* Precipitant Mix 1 (40% *v*/*v* PEG 500 MME; 20% *w*/*v* PEG 20000). The crystals were flash frozen in the crystallization solution and used for data collection at the Diamond Light Source -i03 beamline for *Dr*DyP and the Petra-P13 beamline for *Dr*DyPM190G. 

X-ray diffraction data were produced, illuminating crystals with synchrotron radiation under a nitrogen stream at 100 K. *Dr*DyP data were collected at P13 beamline of PETRA III synchrotron, operated by EMBL, Hamburg (Germany), and *Dr*DyPM190G at i03 beamline of Diamond Light Source (Oxfordshire, UK). Diffraction data were indexed, integrated, and scaled with program XDS and autoPROC (Staraniso), respectively [36,37].

*Dr*DyP and *Dr*DyPM190G mutant crystals contain one and two molecules per asymmetric unit, corresponding to 52.6% and 45.4% of solvent content, respectively [38,39]. *Dr*DyPM190G structure was solved with molecular replacement using program PHASER [40] within the PHENIX suite [40] and the coordinates of *Auricularia Auricula-judae* Dyp as a searching model (PDB code: 4W7L). *Dr*DyP structure was solved with PHASER [40] using *Dr*DyPM190G as a searching model, showing 8.7 and 63.7 TFZ parameters do confirm the corrections of the models, respectively [41]. Crystallographic refinement was carried out with PHENIX.REFINE [41,42] and manual model building with Coot [43] upon examination of 2m|Fo|-D|Fc| and m|Fo|-D|Fc| electron-density maps. Water molecules were assigned automatically from m|Fo|-D|Fc| difference map peaks, when hydrogen bonding acceptors or donors were within 2.45–3.40 Å distance, but only kept if their a.d.p.s refined to values lower than 60 Å^2^. Iterative cycles of refinement and model building proceeded until convergence of R_free_ and R_work_ values, and a final cycle of refinement was produced including all available data. Stereochemistry of the refined structures was analyzed with MOLPROBITY [44].

The data collection parameters, processing, and refinement statistics are listed in Table 1. The structures have been deposited in the PDB databank with the PDB IDs 8RE2 (*Dr*DyP) and 8RE3 (*Dr*DyPM190G).

## Figures and Tables

**Figure 1 molecules-29-00358-f001:**
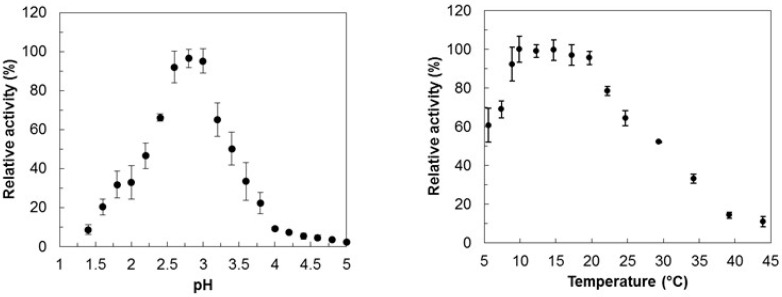
pH-activity profile (**left panel**) and temperature-optimum profile (**right panel**) of *Dr*DyP using ABTS and H_2_O_2_ as substrates.

**Figure 2 molecules-29-00358-f002:**
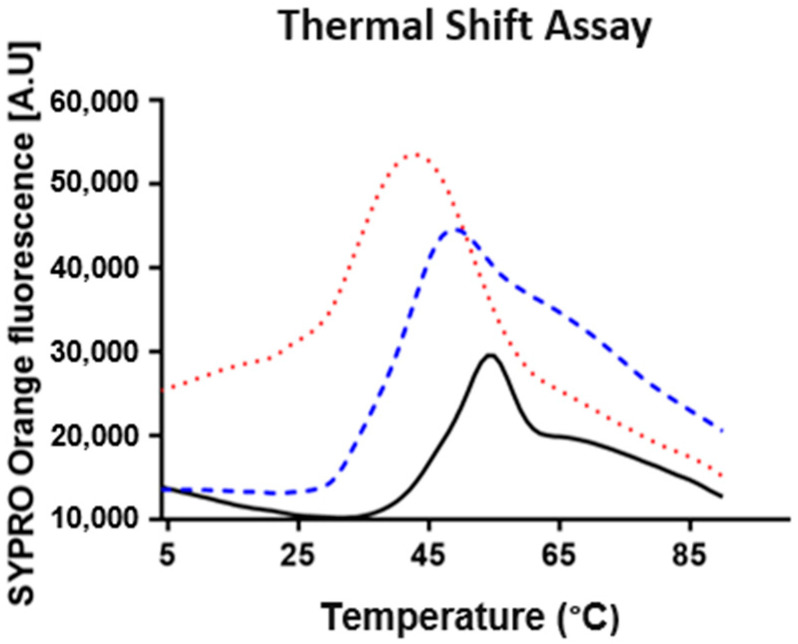
Buffer-stability analysis. The *Dr*DyP thermal-shift assay for 0.1 M Citrate-Phosphate buffer pH 3.0, 0.1 M Britton–Robison buffer pH 3.0, and 0.05 M Tris-HCl pH 7.5/0.150 M is represented as a red dotted line, blue dashed line, and black solid line, respectively. The analysis was acquired from 4 to 90 °C.

**Figure 3 molecules-29-00358-f003:**
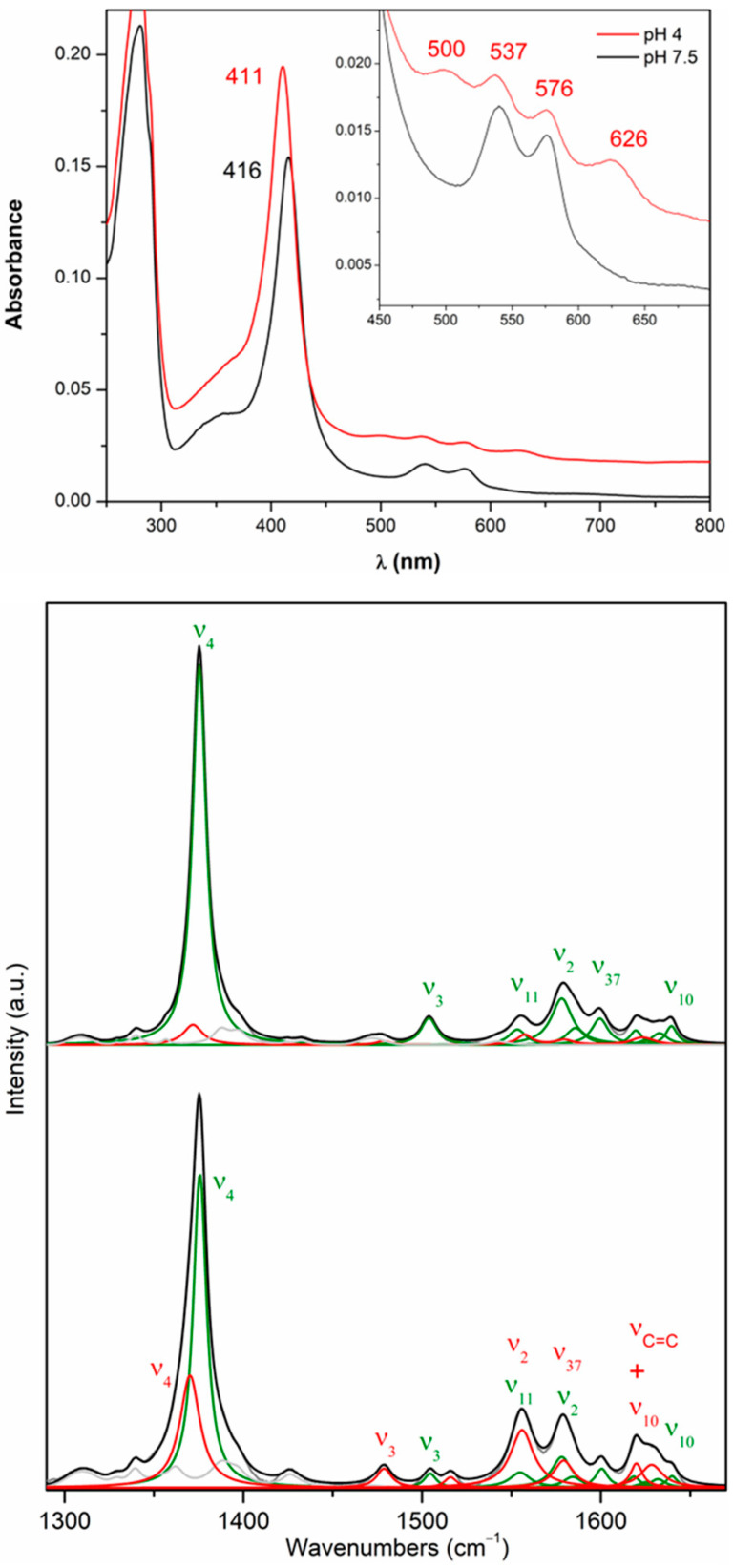
UV-Visible and RR spectra of *Dr*DyPM190G. **Top panel**: UV-Vis spectra measured in BR buffer solutions adjusted to pH 7.5 (black) and 4.0 (red). The baseline of the spectrum acquired at pH 4 was vertically shifted by adding a constant value. **Middle** and **bottom panels**: Component analysis of RR spectra measured at pH 7.5 and 4.0, respectively. 6cLS and 6cHS populations are represented in green and red. Overall fit to the experimental data (gray) is depicted in black. Non-assigned bands are shown in light gray. Spectra were acquired at room temperature with 413 nm laser excitation in BR buffer solutions adjusted to different pH values.

**Figure 4 molecules-29-00358-f004:**
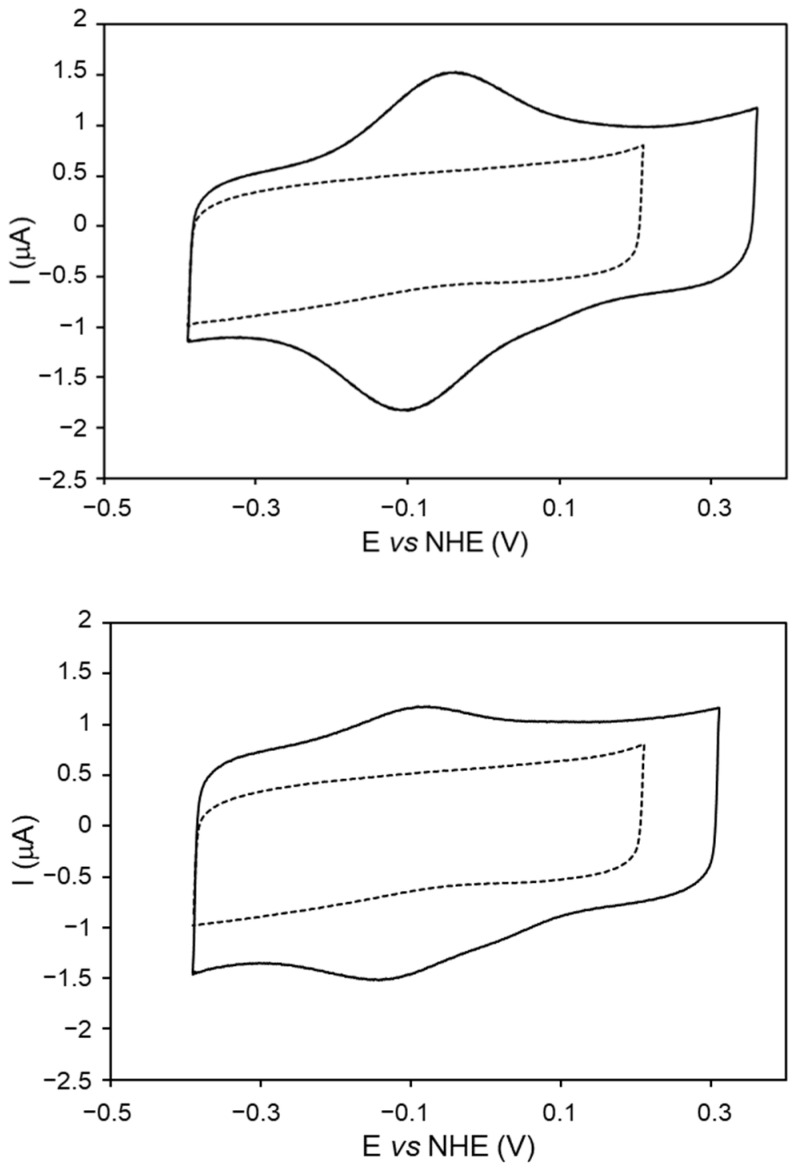
Cyclic voltammograms of *Dr*DyP (**top panel**) and *Dr*DyPM190G (**bottom panel**) adsorbed on pyrolytic graphite electrodes. The control electrode, prepared with protein buffer, is represented by the dashed line. Measurements performed in 50 mM KCl in BR buffer, pH 7.0, and 50 mV/s scan rate.

**Figure 5 molecules-29-00358-f005:**
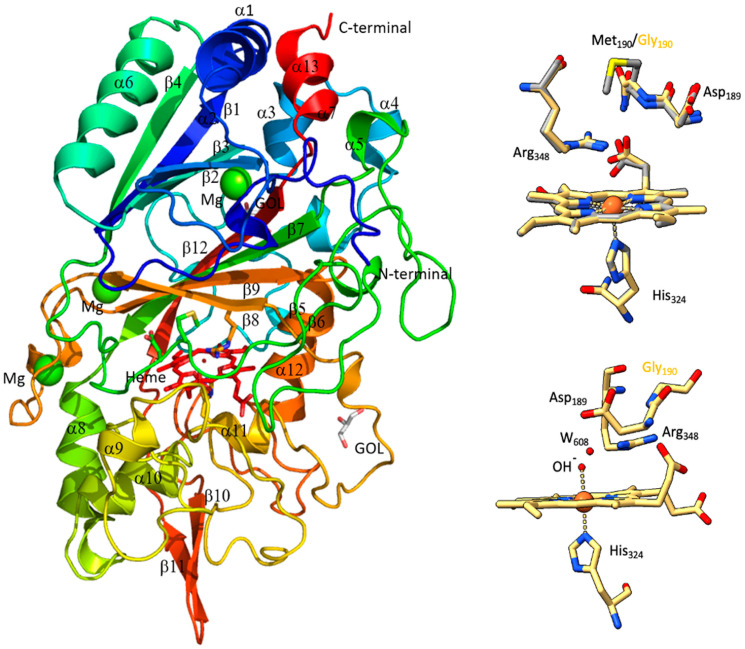
X-ray structures of *Dr*DyP and *Dr*DyPM190G. The overall structure of *Dr*DyP (**left panel**), superimposition of hemes and catalytic residues of *Dr*DyP (coloured) and *Dr*DyPM190G (grey) (**top right panel**), and distances between the hydroxyl group and the water molecule on the distal side of heme iron (2.2 Å to OH), Asp189 (3.8 Å to dH_2_O) and Arg348 (2.8 Å to OH) of *Dr*DyPM190G (**bottom right panel**). The distance between the hydroxyl group and the water molecule is 2.4 Å. The Figures were prepared with PyMOL Version 1.3 (https://pymol.org/2/, 1 November 2023).

**Table 1 molecules-29-00358-t001:** Crystallographic parameters and structure-refinement statistics.

DyP	*Dr*DyPM190G	*Dr*DyP
Beamline	Petra-P13	DLS-I03
Space group ^a^	P2_1_2_1_2_1_	P3_2_
Wavelength (Å)	0.99999	0.97625
Resolution (Å) ^b^	74.74–1.51 (1.65–1.51)	55.54–2.20 (2.33–2.20)
Unit-cell parameters (Å)	a = 82.090 b = 97.70 c = 116.04	a = b = 64.130 c = 111.330
Data processing	AutoPROC/STARANISO	XDS
Resolution limits of ellipsoid fitted to resolution cut-off surface (Å)	2.03, 1.49, 1.60	n/a
Resolution, spherical limits (Å)	74.74–1.51 (1.65–1.51)	55.54–2.20 (2.33–2.20)
No. of observations	567,144 (24,552)	104,559 (16,905)
No. of unique reflections	98,273 (4914)	25,853 (4196)
Multiplicity	5.8 (5.0)	4.04 (4.06)
*<I>*/*<σ(I)>*	11.3 (1.7)	10.96 (0.97)
%CC ^1/2^	99.7 (78.4)	99.8 (50.8)
% R_merg_	7.1 (84.9)	6.6 (108.1)
% R_meas_	7.8 (94.6)	7.6 (124.4)
% R_pim_	3.2 (40.8)	3.52 (34.7)
Data completeness (%), sperical (%)	67.2 (14.2)	99.5 (99.1)
Data completeness, ellipsoidal (%)	93.4 (57.4)	n/a
Solvent contents (%)	45.8	36.33
Wilson B factor (Å^2^)	17.99	55.82
**Refinement**		
Resolution range	74.74–1.51 (1.53–1.51)	39.31–2.20 (2.26–2.20)
No. protein molecules in *a.u.*	2	1
No. of residues per chain (total)	444; 444 (888)	441 (441)
No. of solvent waters	653	50
No. of Haem groups per chain	1, 1 (2)	1
No. of glycerol molecules per chain	-	2
No. of Ca^2+^ ions	1, 1 (2)	-
No. of Mg^2+^ ions	-	3
No. of Cl^−^ ions	1, 1 (2)	-
R_work_	0.167 (0.317)	0.183 (0.360)
R_free_	0.195 (0.187)	0.237 (0.398)
Estimated coordinate error	0.13	0.34
<adp>s of protein chains (Å^2^)	26.2, 24.2	59.09
<adp>s of haem groups per chain (Å^2^)	-	-
<adp>s of solvent waters (Å^2^)	29.3	53.10
**R.m.s.d deviations from ideal values**		
Bonds (Å)	0.010	0.008
Angles (°)	1.050	0.963
**Ramachandran plot analysis: %residues in**		
Favoured regions (%)	97.97	98.18
Allowed regions	1.80	1.82
Disallowed regions	0.23	0.0

Notes: ^a^ Tetragonal and trigonal crystals grown at 277 and 298 K, respectively. ^b^ Values within parentheses correspond to highest-resolution shells.

## Data Availability

Data are contained within the article and Appendix A.

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
