# Peer review of "Biochemical, Biophysical, and Structural Analysis of an Unusual DyP from the Extremophile Deinococcus radiodurans"

_molecules, 2024, doi:10.3390/molecules29020358_

Round 1

Reviewer 1 Report

Comments and Suggestions for Authors

The manuscript by Frade and co-workers "Biochemical, biophysical, and structural analysis of an unusual DyP from the extremophile Deinococcus radiodurans" investigates the WT protein and a variant, which mimicks common DyPs regarding its DyP defining fingerprint motif. The paper is in principle a fine contribution, although there is not a true effect of the Met to Gly exchange. Unfortunately the paper ignores a big part of the relevant literature and also more experiments are needed in order to make this manuscript fit for publication. For example, in-depth spectroscopic studies have to be performed (will be listed in detail below). Further the analysis of the crystal structure concerning the distal ligands are flawed. A single oxygen molecule does not bind on ferric heme. In case a dioxygen would bind, a ferrous state would have to be present, which consequently would require more investigations (see details below).

In detail (minor and major) some of the aspects that have to be addressed and clarified in order to publish this work are:

- Abstract line 9: recently is a rather vague explanation. I would not consider a discovery from more than 20 years ago to be "recently".

- Abstract lines 22-26: This summary of the results, referring to oxygen molecules (!) as ligands cannot be true and this part of the Abstract needs to be rewritten after revision of the rest of the manuscript. Two oxygen molecules mean two O2 molecules, which are not even present in the data and figures of the authors. There two oxygen atoms are present, which most probably are H2O molecules, as the protons are not visible at these resolutions (they only get visible at atomic resolution at around 0.9 Angstroms).

- line 45: please pay attention to correct syntax: kcat/KM

- line 61: heme b

- lines 62 and following: the introduction of the GXXDG motif should be expanded by the fact that the distal aspartate is part of the loop connecting the N-terminal and C-terminal domain of one DyP subunit (https://pubmed.ncbi.nlm.nih.gov/32891739/), which is of special importance in these enzyme classes.

- lines 78 and following (part 1): please pay attention to correct syntax. The oxidation state of transition metals present in complexes (as iron in porphyrins) has to be given in roman numbers (e.g. Fe(III), or Fe(IV)), the writing of e.g. Fe2+ refers to free ions (https://iubmb.qmul.ac.uk/)

- lines 78 and following (part 2): references 12 and 13 are not really appropriate to discuss the reaction mechanism via Compound I. This has already been shown directly already for other DyPs and need to be cited correctly (https://pubmed.ncbi.nlm.nih.gov/30072383/; https://pubmed.ncbi.nlm.nih.gov/32780931/; https://pubmed.ncbi.nlm.nih.gov/36366763/).

- Figure 1 legend: the term "temperature pH profile" is misleading. In the right panel you show a temperature profile at the pH optimum (pH 2.8 according to the materials section)

- Paragraph 2.2.: please pay attention to correct syntax: TM

- Paragraph 2.3.: (i) spectroscopic analysis of the enzyme's of interest is not complete. It would be highly necessary to determine the alkaline transition between the high-spin and low-spin state. This can be followed also by monitoring the spectral changes in the UV-vis absorption spectra. (ii) In Figure 3 (top pane) the UV vis spectra at pH 4 and 7.5 are shown and a clear difference is observable. In case the baseline of the spectrum at pH 4 is shifted, it should be stated in the figure legend, otherwise the measurements should be repeated for better baseline quality.  (iii) please show the spectrum which is referred to (spectrum not shown, line 163). (iv) The resonance Raman spectra should be obtained using different excitation wavelengths (not only 413 nm). Excitation of the samples at pH 4 should be also performed e.g. using the 406 excitation to enhance the high-spin modes. (v) The origin of the low-spin should be identified, it is possible to determine between different ligands, by not only observing the high wavenumber region, but also the low wavenumber region. Further O18 labeling might confirm or exclude that OH- is responsible for the low-spin. A hydroxide ligand would fit to the bands in the visible region of the absorption spectrum (close to 545 and 575 nm). (vi) Cyanide acts as a low-spin ligand, but causes completely different spectral transitions in the UV-vis absorption spectrum (usually one major peak around 540 nm and a shoulder at higher wavelengths, and a Soret band at around 422 nm), it is not a good example to make a case for the present low-spin ligand at physiological pH (as was performed in reference 16, figure 7).

- line 217 and following: The literature search on reduction potential is incomplete (https://pubmed.ncbi.nlm.nih.gov/31325671/)

- line 240: please cite all relevant literature according to flipped heme orientations (https://pubmed.ncbi.nlm.nih.gov/32034988/; https://pubmed.ncbi.nlm.nih.gov/12837059/,... )

- line 241 and following: please refer to more than family V of DyP, when discussing structural elements. Most important re-evaluate your interpretation of the distal heme side. It has been shown that there are up to two water molecules sitting, depending of the redox state of the heme iron. Depending on the data acquisition method (using low-dose) (https://pubmed.ncbi.nlm.nih.gov/32723869/; https://pubmed.ncbi.nlm.nih.gov/34477969/; https://pubmed.ncbi.nlm.nih.gov/32780931/; https://pubmed.ncbi.nlm.nih.gov/31942590/). In the structure of reference 25 (7d8m) the lower oxygen atom of the modeled O2 in the distal side is at a distance of 4 Angstroms from the porphyrin iron. This is not to be considered a coordinating ligand. In reference 26 the distance is 3.4 Angstrom, and nevertheless in order to have an O2 binding the iron would have to be originally in a ferrous state. The other possibility of Compound III is also rather unlikely, since the structure was not obtained during turnover. The most likely situation is that actually hydroxide or water molecules are present on the distal side. As stated before the description (also in lines 257 and 272) of two (!) oxygen molecules (!) is definitely incorrect, in that case there should be oxygen atoms visible, to prove this more spectroscopic studies also using EPR and extended resonance Raman would have to be performed. But such studies would most probably not lead to anything, as there is no dioxygen in the present structures.

- line 384: please pay attention to correct syntax: 0.3 M

- line 393: please pay attention to correct syntax: DyP

- line 406 and Table 1: please pay attention to correct syntax: Rfree Rwork etc.

- Table 1: The intensity of sigma for the highest resolution shell is below 1, so the noise is obviously more intense than any signal, maybe a higher resolution cut-off has to be considered, also because the data completeness for the highest resolution shell is very low for the M190G variant (66.6 %), indicating the same

- please provide the pdb-validation reports (.pdf) for the next submission

Reviewer 2 Report

Comments and Suggestions for Authors

In the current manuscript, the authors have reported the biochemical, biophysical, and structural characterization of dye-decolorizing peroxidases. The experiments are well-designed, and the results corroborate with the conclusions. The manuscript is acceptable for publication; however, the authors must address the following queries.

1.     In Figure 1, the authors have represented the pH and temperature dependence of the activity of DrDyP. They have performed the experiments for pH dependence using ABTS and the experiment for temperature dependence using H2O2. The authors should report the pH as well as temperature dependence using both substrates. 

2.     How did the authors decide on the concentrations of H2O2 and ABTS to be used in the assays? Are the relative activities estimated from the kcat values at different pHs and temperatures. If not, the authors should estimate the Km and kcat values at different temperatures and pHs. 

Comments on the Quality of English Language

Listed below are a few corrections for the English language in the manuscript.

a.     Lines 82-83: Reword the last part of the sentence as ‘……with hydrogen peroxide as electron acceptor for cleavage….’

b.     Lines 89 and 90: Correct ‘intermediaries’ to ‘intermediates’.

c.     Line 91: Correct ‘spectroscopies’ to ‘spectroscopy’

d.  Lines 91-93: The sentence is not complete. The authors must describe what the important tools are.

e.     Line 203: Change 'catalytic' to 'catalytically'.

Round 2

Reviewer 1 Report

Comments and Suggestions for Authors

The manuscript is now significantly improved. There is only one point still unclear that definitely has to be resolved prior to publication.

This concerns the data collection statistics in Table 1 of the 8RE3 structure and the data that can be found in the validation report. The authors stated that the number of 66% completeness was by accident incorrectly added in table 1. Now the number is 93.4% in the table. Nevertheless, in section 4 of the validation report (page 6) the number is still 66.5%.

This needs to be clarified.
